# Merging Experts into One:
# Improving Computational Efficiency of Mixture of Experts

**Shwai He**[1] **Run-Ze Fan**[3] **Liang Ding**[2*] **Li Shen**[4] **Tianyi Zhou**[1*] **Dacheng Tao**[2]

[1]University of Maryland, College Park   [2]The University of Sydney
[3]University of Chinese Academy of Sciences   [4]JD Explore Academy

shwaihe@umd.edu, liangding.liam@gmail.com, tianyi@umd.edu

## Abstract

Scaling the size of language models usually leads to remarkable advancements in NLP tasks. But it often comes with a price of growing computational cost. Although a sparse Mixture of Experts (MoE) can reduce the cost by activating a small subset of parameters (e.g., one expert) for each input, its computation escalates significantly if increasing the number of activated experts, limiting its practical utility. Can we retain the advantages of adding more experts without substantially increasing the computational costs? In this paper, we first demonstrate the superiority of selecting multiple experts and then propose a computation-efficient approach called `Merging Experts into One` (MEO), which reduces the computation cost to that of a single expert. Extensive experiments show that MEO significantly improves computational efficiency, e.g., FLOPS drops from 72.0G of vanilla MoE to 28.9G (MEO). Moreover, we propose a token-level attention block that further enhances the efficiency and performance of token-level MEO, e.g., 83.3% (MEO) vs. 82.6% (vanilla MoE) average score on the GLUE benchmark. Our code will be released upon acceptance. Code will be released at: https://github.com/Shwai-He/MEO.

## 1 Introduction

Scaling language models has achieved promising progress in the field of NLP (Brown et al., 2020; OpenAI, 2023). To further increase the model size under a computational budget, sparsely activated networks (Du et al., 2022; Artetxe et al., 2022) only employ a few parameters for each input. A widely studied approach is the Mixture-of-Experts (MoE, Shazeer et al., 2017), which trains multiple expert networks but only selects a subset of them for a specific input (Jacobs et al., 1991; Jordan and Jacobs, 1994). Compared to dense networks of the same model size, MoE effectively reduces computational costs.

---
*Corresponding author

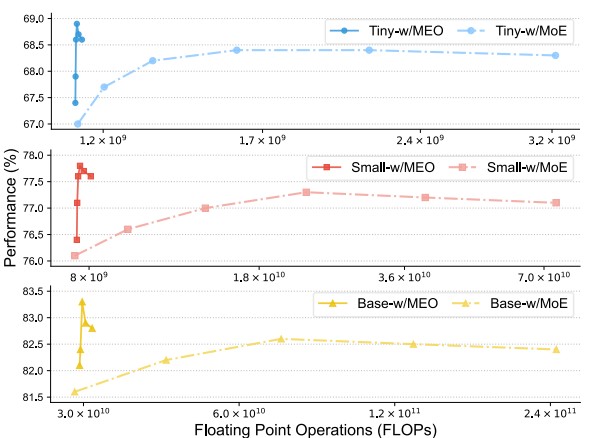

Figure 1: **Performance vs. FLOPs of MoE and MEO** at the token level when different numbers of experts (i.e., 1, 2, 4, 8, 16, 32) are selected. We take three different sizes of BERT as the expert model.

Although increasing the experts selected for each input can improve the representation diversity (Yang et al., 2019) and downstream task performance (Shazeer et al., 2017; Yang et al., 2019), it usually comes with a price of significantly growing computational cost. Our empirical study (Figure 1 and Table 1) verifies the Pros and Cons (*superior performance* vs. *high computational cost*) of selecting multiple experts at MoE inference. Hence, to retain the advantage of MoE on computational efficiency, existing work mainly selects only one expert per input in applications or experiments (Fedus et al., 2021), which inevitably compromises the performance.

Our work aims to improve the computational efficiency of MoE inference with multiple experts selected, for greatly rejuvenating the compromised performance. The computation involved in MoE primarily consists of the inference on each selected expert and the summation of their outputs, with the former dominating the cost. Hence, the cost linearly grows with the number of selected experts. To overcome the computational bottleneck, we instead propose **Merging Experts into One** (MEO),

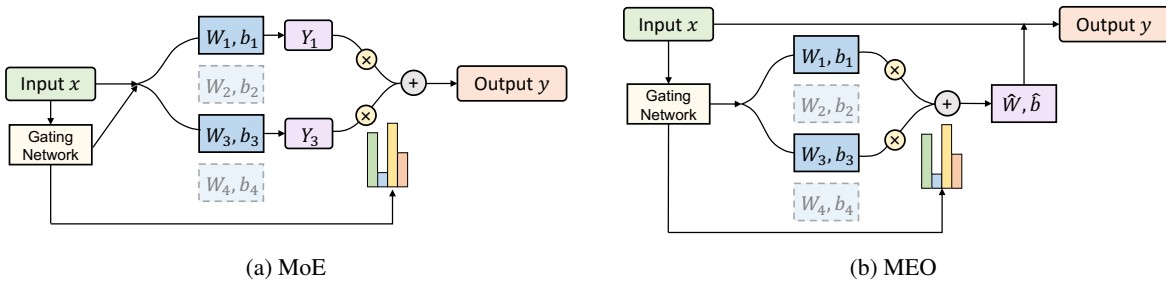

|                | (a) MoE | | (b) MEO |

Figure 2: **The diagrams of (a) MoE and (b) our proposed MEO**, with a case of $m = 2$ experts are selected. MoE linearly combines the outputs from experts, while MEO first merges experts into one and then computes the input.

which alters the calculation order of the two operations, i.e., first merging the parameters of the selected experts into one expert followed by inference on the merged expert. Since the parameter merging only requires summation, an MEO layer (approximately) only consumes the computation of single-expert inference, no matter how many experts are selected. This leads to a nearly constant inference cost when scaling up the model capacity (i.e., the number of selected experts) to improve the performance.

MEO can be applied as a drop-in replacement for MoE, which has been deployed at various levels, e.g., selecting experts for each token (Shazeer et al., 2017), each sequence (Ye et al., 2022), each task (Kudugunta et al., 2021), etc. On the sequence/task level, our empirical studies demonstrate that replacing MoE with MEO significantly improves computational efficiency, e.g., reducing FLOPs from 72.0G to 28.9G, without hurting the performance. In addition, we propose a token-level attention mechanism that further enhances the efficiency and performance, e.g., from 82.6% (MoE) to 83.3% (MEO) on BERT-Base (Figure 1).

## 2 Methodology

**Review of Mixture of Experts.** Given a token $x_i$ in the input sequence $x \in \mathbb{R}^{s \times d}$, MoE selects $m$ experts from $n$ ($m \leq n$) experts $(E_1, \ldots, E_n)$ based on a gating network. We denote $\mathcal{G}$ as the gating scores and $\mathcal{T}$ as the indices of selected experts. MoE linearly combines the outputs of selected experts:

$$y_i = \sum_{k \in \mathcal{T}} \mathcal{G}_k(x_i) \cdot E_k(x_i). \quad (1)$$

MoE performs at various levels, e.g., token, sequence, and task, where MoE selects experts based on a single token, input sequence, or task embedding (or task ids):

$$\mathcal{G}(x_i) = \begin{cases} \text{GATE}(x_i), & \text{Token-level} \\ \text{GATE}(\frac{1}{s} \sum_{i=1}^{s} x_i), & \text{Sequence-level} \\ \text{GATE}(task\_ids), & \text{Task-level} \end{cases} \quad (2)$$

where "GATE" denotes the gating function.

Table 1: **Effects of the number of selected experts** on performance. The best results are **bold**.

| $m$ | #FLOPs. | SST-2 | STSB | MNLI | QNLI | Avg. |
|---|---|---|---|---|---|---|
| 1  | 7.5G  | 87.1 | 86.1 | 77.8 | 85.8 | 84.2 |
| 2  | 9.6G  | 87.9 | 86.8 | 78.2 | 86.2 | 84.8 |
| 4  | 13.9G | 88.2 | 87.1 | 78.3 | 86.4 | 85.0 |
| 8  | 22.5G | 88.3 | **87.7** | **79.1** | **86.8** | **85.5** |
| 16 | 39.7G | **88.4** | 87.5 | 78.8 | 86.6 | 85.3 |
| 32 | 74.1G | 88.2 | 87.6 | 78.6 | 86.3 | 85.2 |

**Motivation.** While many predominant MoE models tend to select the top-1 expert (Fedus et al., 2021), selecting multiple experts has the potential of boosting the representation power (Chen et al., 2020; Yang et al., 2019). Empirically, we conduct preliminary experiments on the BERT-Small (Bhargava et al., 2021) to verify it.

In Table 1, it is evident that selecting multiple experts contributes to better performance. Even though selecting excessive experts is suboptimal as it introduces the interference between experts that hinders the performance (Mustafa et al., 2022; Zhu et al., 2022), our preliminary experiments necessitates the selection of multiple experts.

However, selecting more experts leads to a substantial increase in FLOPs (e.g., 74.1G v.s. 7.5G when increasing $m$ from 1 to 32). This phenomenon urges us to reflect *whether there exists an efficient approach to achieve both high performance and computational efficiency*. Our goal is to ensure consistent computational cost, regardless of the number of selected experts.

**Merging Experts into One.** The computation cost of MoE primarily involves the computation of individual experts (i.e., $\sum_{k \in \mathcal{T}} O(E_k)$) and the mixture of outputs from experts (i.e., $O(\mathcal{G})$ and $O(\sum_{k \in \mathcal{T}} \mathcal{G}_k \cdot E_k)$). Notably, the computation of individual experts plays a dominant role, with even the cost of a single expert being significantly outweighing that of the mixture:

$$O(E_k) \gg O(\mathcal{G}) + O(\sum_{k \in \mathcal{T}} \mathcal{G}_k \cdot E_k), \quad (3)$$

where $O(\cdot)$ measures the computational cost.

On the other hand, as the number of selected experts $m$ increases, the term $\sum_{k \in \mathcal{T}} O(E_k)$ experiences a substantial increase, whereas the increase in $O(\sum_{k \in \mathcal{T}} \mathcal{G}_k \cdot E_k)$ is marginal. Therefore, it is essential to address the growing trend of $\sum_{k \in \mathcal{T}} O(E_k)$ to enhance computational efficiency.

As illustrated in Figure 2, we propose the method called Merging Experts into One (MEO), where the key idea is to leverage the gating scores to aggregate the parameters of the selected experts (which is akin to the simple weighted model fusion mechanism (Li et al., 2023)):

$$\hat{W}_i = \sum_{k \in \mathcal{T}} \mathcal{G}_k(x_i) \cdot W_k, \hat{b}_i = \sum_{k \in \mathcal{T}} \mathcal{G}_k(x_i) \cdot b_k, \quad (4)$$

where $W_k, b_k$ represent the weight and bias of the $k$-th expert, while $\hat{W}_i, \hat{b}_i$ are the aggregated weight and bias for $x_i$. The output of MEO is given by:

$$y_i = \sigma(\hat{W}_i x_i + \hat{b}_i), \quad (5)$$

where $\sigma$ represents the activation function.

The computation cost of MEO primarily consists of $O(\sigma(\hat{W}_i x_i + \hat{b}_i))$, $O(\sum_{k \in \mathcal{T}} \mathcal{G}_k \cdot W_k)$, $O(\sum_{k \in \mathcal{T}} \mathcal{G}_k \cdot b_k)$, and $O(\mathcal{G})$. Among them, $O(\sigma(\hat{W}_i x_i + \hat{b}_i))$ is the dominant factor. It is worth noting that $O(\sigma(\hat{W}_i x_i + \hat{b}_i))$ is equivalent to the computation cost of a fully connected network and independent of the number of selected experts. Therefore, MEO compresses computation costs significantly.

**MEO at Different Levels.** In the case of sequence and task level MEO, all tokens within a sequence share the same gating scores, as well as the aggregated parameters $\hat{W}$ and $\hat{b}$ [1]. This property allows for easy adoption of MEO at these levels.

---

[1] we omit subscripts of $\hat{W}$ and $\hat{b}$ at the sequence and task level given each token shares the same aggregated parameters.

However, when directly applying MEO at the token level, the situation is different. Since the gating scores of each token within a sequence are unique, the straightforward usage of MEO would require the aggregation of multiple sets of weights and biases, resulting in increased deployment cost. Therefore, we refine and enhance the framework of token-level MEO specifically.

**Token-Level MEO.** Our proposed token-level MEO aims to incorporate token-level information with minimal extra computational cost. Specifically, the expert selection is performed at the sequence level, thereby preserving context information and eliminating the necessity of aggregating multiple weights and biases for individual tokens. To capture the identification of each token, we leverage the token attention mechanism inspired by Houlsby et al. (2019); Li et al. (2021).

Specifically, given the input sequence $x \in \mathbb{R}^{s \times d}$, we employ a specialized bottleneck block, inspired by adapter-like structures (Houlsby et al., 2019; Pfeiffer et al., 2021). The bottleneck layer incorporates down-projection weights $\boldsymbol{W}_{down} \in \mathbb{R}^{d \times \frac{d}{r}}$, an activation function $f$ and up-projection weights $\boldsymbol{W}_{up} \in \mathbb{R}^{\frac{d}{r} \times d}$, with reduce factor $r = 64$ that ensures low extra computational cost. By operating on each token individually, the bottleneck applies token-level attention to the input sequence $x$:

$$x \leftarrow x + f(x\boldsymbol{W}_{down})\boldsymbol{W}_{up}. \quad (6)$$

With the inclusion of token identification in the updated input, MEO performs aggregation of $\hat{W}$ and $\hat{b}$ through sequence-level expert selection. Subsequently, these aggregated $\hat{W}$ and $\hat{b}$ are used to compute the output in conjunction with the input.

## 3 Empirical Evaluation

**Experimental Setup.** Experiments were conducted on Four widely-used benchmarks, spanning understanding and generation tasks: (1) GLUE (Wang et al., 2019), containing understanding tasks like natural language inference, sentiment analysis, and sentence similarity evaluation; (2) XSum (Narayan et al., 2018), a summarization dataset where the models are required to generate a short summary for a given article; (3) WikiText-2 (Merity et al., 2016), a collection of over 100 million tokens extracted from the set of verified Good and Featured articles on Wikipedia where the models are utilized to generate the next tokens; (4)

Table 2: **Empirical results for MEO and MoE in task-level ($task$) and sequence-level ($seq$).** We also report the performance of vanilla feedforward layers ("Vanilla") as a reference. The shown results are the averaged score for 5 runs. The best results are **bold**. ✶ indicates the method with the fewest the fewer FLOPs ("Vanilla" is not included).

| Method | #FLOPs. | CoLA | SST-2 | MRPC | STS-B | QQP | MNLI | QNLI | RTE | Avg |
|---|---|---|---|---|---|---|---|---|---|---|
| Vanilla | 28.5G | 54.6 | 91.1 | 84.6 | 85.8 | 90.2 | 80.6 | 90.4 | 66.4 | 80.5 |
| $MoE_{task}$ | 72.0G | 58.5 | 91.3 | 85.8 | 89.2 | 90.5 | 82.7 | 90.5 | 69.3 | 82.2 |
| $MEO_{task}$ | ✶28.9G | 59.1 | 91.2 | 85.5 | 89.3 | 90.4 | 83.0 | 90.9 | 68.9 | 82.3 |
| $MoE_{seq}$ | 72.0G | 59.8 | 91.5 | **86.5** | **89.5** | 90.6 | 83.4 | 90.7 | **70.4** | 82.8 |
| $MEO_{seq}$ | ✶28.9G | **60.1** | **91.9** | 86.3 | 89.4 | **90.7** | **83.7** | **91.2** | 70.3 | **83.0** |

SQuAD v1.1 (Rajpurkar et al., 2016), a pair-wise dataset for questions and Wikipedia paragraphs where models select the answer span to the question from the paragraph.

We follow Zhong et al. (2022a,b); He et al. (2023a) to conduct experiments on the widely-used GLUE benchmark, containing understanding tasks like natural language inference, sentiment analysis, sentence similarity evaluation, etc. We use Adam (Kingma and Ba, 2015) as the optimizer with $\beta_1, \beta_2 = 0.9, 0.98$. For regularization, we set the weight decay as 0.1 and grid-search the learning rate from {1e-5, 5e-5, 1e-4, 5e-4}, where we warm up the learning rate in the first 10% steps (of the total training steps). For different data scales, we grid-search the training epoch and batch size from {5, 10, 15, 20}, and {8, 16, 32, 64}, respectively. The maximum length is 128 for GLUE, 1024 for WikiText, and 384 for SQuAD. For XSum, we set the max length of source articles to be 512 and the max length of the target summary to be 128. We follow previous works (Phang et al., 2018; Lee et al., 2020; Dodge et al., 2020; Wang et al., 2022; He et al., 2023b) to fine-tune the pretrained language models, e.g. BERT (Devlin et al., 2019), on the downstream training set and report results using the last checkpoint.

**Main Results.** Following Shazeer et al. (2017); Gao et al. (2022), we conduct experiments on BERT-Base (Devlin et al., 2019) and replace feedforward layers ("Vanilla") with MoE or MEO, with the setting $m = 4$ and $n = 16$. In Table 2, we carefully compare our proposed MEO with MoE at task and sequence levels, in terms of computational efficiency and performance. Compared to MoE, MEO significantly reduces the computation cost while achieving comparable performance. Specifically, compared to vanilla feed-forward layers, the Floating Point Operations (FLOPs) of MEO only increase marginally (i.e., about 1%), while MoE multiplies the FLOPs about 2.53 times.

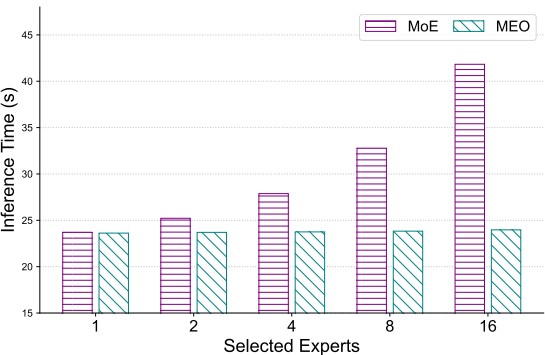

Figure 3: **Comparison of inference time between MoE and MEO** under a series of different numbers of selected experts (i.e., 1, 2, 4, 8, 16).

**Analysis of Reduced Computation.** Compared to a fully connected layer, MEO only introduces computation in gating network $O(\mathcal{G}(x))$ and merging experts (i.e., $O(\sum_{k \in \mathcal{T}} \mathcal{G}_k \cdot W_k)$ and $O(\sum_{k \in \mathcal{T}} \mathcal{G}_k \cdot b_k)$). The additional computation is minimal compared to that of individual experts.

In practice, we use eight NVIDIA V100 Tensor Core GPUs to measure the inference time of MEO and MoE on BERT-Base when selecting different numbers of experts (i.e., $n = 1, 2, 4, 8, 16$). Inference time is calculated by the total running time on the MNLI validation dataset with batch size 16. According to Figure 3, as the number of selected experts increases, the inference time of MEO is relatively consistent, while MoE exhibits a significantly increased inference time. This highlights the advantage of MEO in computational efficiency, which becomes even more pronounced as the number of selected experts grows.

Table 3: **Comparison between MEO and MoE with different activation function usage** (i.e., activation function within ($in$) and outside ($out$) experts).

| Method | FLOPs | SST-2 | QQP | MNLI | QNLI | Avg. |
|---|---|---|---|---|---|---|
| Vanilla | 7.5G | 86.9 | 89.1 | 77.2 | 85.2 | 84.6 |
| $MoE_{in}$ | 22.6G | 87.9 | 89.4 | 77.8 | 85.7 | 85.2 |
| $MoE_{out}$ | 22.5G | 87.6 | 89.2 | 78.0 | 85.6 | 85.1 |
| MEO | ✶7.7G | **88.1** | **89.7** | **78.2** | **86.2** | **85.6** |

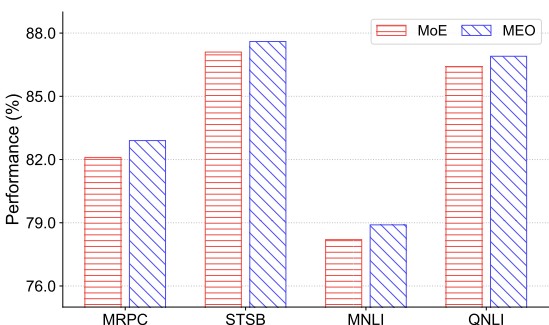

Figure 4: **Performance comparison between token level MoE and MEO**, where we take BERT-Small as the backbone with the setting $m = 8$ and $n = 32$.

**Analysis of Activation Function.** In many cases where an expert $E_i$ represents a linear layer without a nonlinear activation function, the output of MEO $(y_i = \sigma(\hat{W}_i x_i + \hat{b}_i))$ is equivalent to that of MoE $(y_i = \sigma(\sum_{k \in \mathcal{T}} \mathcal{G}_k \cdot (W_k x_i + b_k)))$. However, if the expert $E_i$ involves an activation function, the output of MoE is $y_i = \sum_{k \in \mathcal{T}} (\mathcal{G}_k \cdot \sigma(W_k x_i + b_k))$, which leads to differences in outputs and potentially in performance. As depicted in Figure 3, we compare MEO with MoE with different usage of activation, where we consider two scenarios: activation function within or outside experts. The results demonstrate that the performance gap between the two scenarios is minimal and indicates the effectiveness of MEO in handling expert networks that incorporate activation functions.

Table 4: **Performance of Token-Level MEO**, where we take BERT-Large as the backbone with the setting $m = 2$ and $n = 8$.

| Method | FLOPs | SST-2 | MRPC | STSB | QNLI | Avg. |
|--------|-------|-------|------|------|------|------|
| Vanilla | 87.2G | 93.2 | 86.8 | 89.1 | 91.8 | 90.2 |
| MoE | 139.0G | 93.7 | 87.2 | 89.7 | 92.2 | 90.7 |
| MEO | ✶91.2G | **94.1** | **87.5** | **89.8** | **92.4** | **91.0** |

**Effectiveness of Token-Level MEO.** For MEO at the token-level MEO, we have incorporated token-level attention blocks. To assess the deployment cost of newly added blocks, we first calculate the extra parameters and FLOPs, with BERT-Small as the backbone. The extra cost of added blocks is minimal (i.e., 0.6M parameters and 0.15 GFLOPs). Furthermore, in Figure 4, we present a performance comparison between token level MEO and MoE in four natural language understanding tasks, where MEO outperforms MoE consistently across these tasks, e.g., 78.9% v.s. 78.1% on MNLI. For the average score on the GLUE benchmark, MEO boosts the performance significantly, i.e. 83.3% v.s. 82.6%

on BERT-Base and 77.8% v.s. 77.3% on BERT-Small.

We also implement the token-level MEO on BERT-Large, utilizing 8 experts and selecting 2 experts, resulting in a model with about 1.75 billion parameters. As demonstrated in Table 4, MEO consistently enhances performance across various tasks, e.g., 0.4% improvement in SST-2 when compared to MoE. Notably, the additional computational cost is minimal, with only a 4.0 GFLOPs increase over the Vanilla model. Therefore, token-level MEO proves to be an efficient and effective alternative to token-level MoE.

**Transfer to different architectures and tasks.** Utilizing MEO in BERT architectures enhances computational efficiency and performance, and we further validate the effectiveness of MEO on a wide range of architectures for different tasks. In Table 5, we use BART-Large (Lewis et al., 2020) for XSum (Narayan et al., 2018), GPT-2-Small (Radford et al., 2019) for WikiText (Merity et al., 2016), and T5-Base (Raffel et al., 2020) for SQuAD (Rajpurkar et al., 2016). MEO and MoE are deployed at the token level. Considering the limited computation resource, we set $m = 2$ and $n = 8$ for BART and GPT-2, while $m = 4$ and $n = 16$ are set for T5.

Clearly, MEO outperforms the standard MoE in three tasks, showing its universality in both natural language understanding and generation.

Table 5: **Effectiveness on different architectures and tasks.** XSum, WikiText, and SQuAD are evaluated with ROUGE-2 (R2.), Perplexity (PPL), and Exact Match (EM), respectively.

| Method | XSum | | WikiText | | SQuAD | |
|--------|------|-----|----------|-----|-------|-----|
| | FLOPs | R2. | FLOPs | PPL | FLOPs | EM |
| Vanilla | 369.4G | 21.9 | 295.4G | 21.9 | 90.2G | 81.6 |
| MoE | 576.6G | 22.2 | 412.2G | 21.1 | 221.3G | 82.0 |
| MEO | ✶383.6G | **22.4** | ✶303.2G | **20.9** | ✶93.5G | **82.1** |

## 4 Conclusion

In this work, we systematically investigate the computational cost of the Mixture of Experts. Based on our findings, we propose a drop-in replacement called Merging Experts into One (MEO) to enhance computational efficiency. Additionally, we propose a Token-Level attention mechanism that further boosts performance. Our study empirically indicates the potential to make MEO a golden standard efficient architecture within the NLP community.

# 5 Limitations

Despite the progress we have made, there are still limitations in our work. While our architecture for the mixture of experts demonstrates improved efficiency, there is a need for further exploration in terms of its deployment. Specifically, determining the optimal number of experts in specific layers and selecting different levels of MoEs require additional investigation. We believe that with the implementation of efficient deployment strategies, our method has the potential to become even more competitive.

## Acknowledgements

We are grateful to the anonymous EMNLP reviewers and the area chair for their insightful comments and suggestions.

## Ethics Statement

We take ethical considerations seriously and strictly adhere to the EMNLP Ethics Policy. This paper focuses on the higher efficiency of dynamic networks, e.g., the mixture of experts. Both the datasets and models used in this paper are publicly available and have been widely adopted by researchers. We ensure that the findings and conclusions of this paper are reported accurately and objectively.

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
