# OpenReview forum: "Merging Experts into One: Improving Computational Efficiency of Mixture of Experts"
_EMNLP/2023/Conference — EMNLP 2023 Main_

### Official Review · Reviewer_92Jb · 2023-08-03

**Soundness:** 4

**Excitement:**

4: Strong: This paper deepens the understanding of some phenomenon or lowers the barriers to an existing research direction.

**Missing References:**

AdaMix: Mixture-of-Adaptations for Parameter-efficient Model Tuning -- https://arxiv.org/abs/2205.12410

**Paper Topic And Main Contributions:**

The main topic is improving the computational efficiency of mixture of experts (MoE) models for natural language processing tasks. The key contributions are:

1.	Proposing a new method called Merging Experts into One (MEO) that reduces the computational cost of MoE models to be equivalent to using just a single expert. This is achieved by first merging the parameters of selected experts before computing the input, rather than computing each expert's output separately.
2.	Demonstrating that MEO can be applied at various levels (token, sequence, task) as a drop-in replacement for MoE, while significantly reducing FLOPS. For example, it reduces FLOPs from 72G for MoE to 28.6G for MEO at the sequence level.
3.	Introducing a token-level attention mechanism for token-level MEO that further enhances efficiency and performance. MEO with token attention outperforms vanilla MoE, e.g. 83.3% vs 82.6% on GLUE benchmark with BERT-Base.
4.	Providing extensive experiments that validate the efficiency and effectiveness of MEO compared to MoE. For instance, MEO achieves comparable performance to MoE on GLUE while using far fewer FLOPs.

**Reasons To Accept:**

1.	The paper is clearly written and easy to follow. The method and results are conveyed effectively.
2.	The paper proposes an idea of merging expert parameters before computation to reduce costs. This is a simple but impactful innovation.
3.	The proposed MEO is shown to be effective as a drop-in replacement for MoE at multiple levels (token, sequence, task) with minimal changes. And, the authors further boost efficiency and performance of token-level MEO compared to vanilla token-level MoE inspired token attention.

**Reasons To Reject:**

1.	While GLUE benchmark is used, the experiments are still limited to mostly NLU tasks. Testing MEO on a wider range of tasks could be beneficial.
2.	Only BERT-base model is evaluated. Testing MEO on larger size models could be beneficial.

**Reproducibility:**

4: Could mostly reproduce the results, but there may be some variation because of sample variance or minor variations in their interpretation of the protocol or method.

**Reviewer Confidence:**

5: Positive that my evaluation is correct. I read the paper very carefully and I am very familiar with related work.

---

> ### Author Rebuttal · Authors · 2023-08-29
>
> We appreciate your positive comments and support! We carefully address your concerns as follows:
>
> ## Q1. While GLUE benchmark is used, the experiments are still limited to mostly NLU tasks. Testing MEO on a wider range of tasks could be beneficial.
>
> Thank you for the suggestions! We conducted new experiments on other benchmark tasks beyond the GLUE benchmark, i.e. T5-base on SQuAD, GPT-2 on Wikitext and Bart-large on XSum, the experimental results for token-level MoE and MEO are compared in the tables below.
>
> *BART-large on XSum (max_source_length=512, select 2 experts from 8):*
> | | FLOPs | Rouge-2 |
> | :------ | :------: | :------: |
> | Vanilla | 369.3G | 21.9 |
> | MoE | 576.5G | 22.2 |
> | **MEO** | **369.7G** | **22.4** |
>
> *GPT-2 on Wikitext (max_length=1024, select 2 experts from 8):*
> | | FLOPs | PPL |
> | :------ | :------: | :------: |
> |Vanilla | 279.2G | 21.5 |
> | MoE | 395.9G | 21.1 |
> | **MEO** | **279.3G** | **20.9** |
>
> *T5-base on SQuAD (max_length=384, select 4 experts from 16):*
> | | FLOPs | Exact_Match |
> | :------ | :------: | :------: |
> | Vanilla | 90.3G | 81.6 |
> | MoE | 221.0G | 82.0 |
> | **MEO** | **90.4G** | **82.1** |
>
> Our proposed MEO achieved consistent improvement across different tasks and language models. Moreover, its advantage on computational efficiency still holds when applied to  different mainstream architectures (i.e., Encoder-Only (BERT), Encoder-Decoder (BART and T5), Decoder-Only(GPT-2)), which indicates the universality of our method.
>
> ## Q2. Testing MEO on larger size models could be beneficial.
>
> When applied to 16-expert MEO or MoE, the total number of parameters in BERT-base is about 1 billion. We also conducted experiments on BERT-large with 8 token-level experts (1.75 billion parameters). The new experimental results are reported below:
>
> BERT-large on the GLUE benchmark (max_length=128, select 2 experts from 8):
> | | FLOPs | SST-2 | MRPC | STSB | QNLI	 | Avg. |
> | :------ | :------: | :------: | :------: | :------: | :------: | :------: |
> | Vanilla | 87.2G | 93.2 | 86.8 | 89.1 | 91.8 | 90.2 |
> | MoE | 138.9G | 93.7 | 87.2 | 89.7 | 92.2 | 90.7 |
> | **MEO** | **87.5G** | **94.1** | **87.5** | **89.8** | **92.4** | **91.0** |
>
> When increasing the model size, we also witness the advantages of MEO on both performance and computational efficiency. For different tasks or architectures, please refer to our response to Q1.
>
> ## Q3. Missing References
> Although both our work and AdaMix[1] merge the blocks of language models, our work is fundamentally different from it. First, we focus on enhancing the computational efficiency of MoE, while Aadmix aims at improving the performance of parameter-efficient tuning. Second, the merging operations are different: AdaMix averages the parameters of adapters while our method follows the aggregation process of MoE. That being said, AdaMix’s high-level idea of merging adapters is related. We will cite and discuss AdaMix in the next version.
>
> ## References:
>
> [1] AdaMix: Mixture-of-Adaptations for Parameter-efficient Model Tuning, EMNLP 2022

---

### Official Review · Reviewer_tZMq · 2023-08-05

**Typos Grammar Style And Presentation Improvements:** Line 115 - 74 v 7.4G, should be swapped.
**Soundness:** 4

**Excitement:**

3: Ambivalent: It has merits (e.g., it reports state-of-the-art results, the idea is nice), but there are key weaknesses (e.g., it describes incremental work), and it can significantly benefit from another round of revision. However, I won't object to accepting it if my co-reviewers champion it.

**Paper Topic And Main Contributions:**

The authors propose a new approach for MoE called Merging Experts into One (MEO) which first selects N experts and combines them into a single expert. This method significantly improves computational efficiency (over 2x wrt FLOPS), and outperforms vanilla MoE on GLUE.

**Questions For The Authors:**

- Does this approach work for routing token / seq to FFN (e.g. stacks of linear layers with nonlinearities after each one, without having to have a gating mechanism per linear layer?).
- Why are the inference times for BERT 20+ seconds even for 1 expert? This seems significantly longer than what I'd expect -- what is the batch size?

**Reasons To Accept:**

- Very simple idea that improves efficiency and accuracy of MoE (thought severely limited compared to MoE)
- Overall improvements on GLUE

**Reasons To Reject:**

- The proposed method seems to only work for experts that are essentially an affine transform / linear layer. This is just one interpretation of MoE, and feasibly we can imagine an expert to be any type of network. The method for aggregating experts (e.g. here a weighted sum) only work mathematically if there are no nonlinearities, limiting the usefulness of this approach
- Approach has only been applied to a single model, BERT-Small, it would be good to contain additional datasets outside of GLUE and a more varied treatment of different model architectures and sizes.

**Reproducibility:**

4: Could mostly reproduce the results, but there may be some variation because of sample variance or minor variations in their interpretation of the protocol or method.

**Reviewer Confidence:**

4: Quite sure. I tried to check the important points carefully. It's unlikely, though conceivable, that I missed something that should affect my ratings.

---

> ### Author Rebuttal · Authors · 2023-08-29
>
> Thanks for your insightful comments, which help us further improve our paper. We carefully address your concerns as follows:
>
> ## Q1. The usefulness of our proposed method.
> Most works on MoE recently focus on linear-layer experts [1-3] without nonlinearity and they achieved promising results in practice. So we mainly follow their setting. In Line 236-251, we also discussed how to directly apply our method to nonlinear experts with non-linear activations. The flexible mixture of different networks architecture is out of the main scope of the settings studied in this paper but deserves further exploration in the future. We will add more detailed discussion in the next version.
>
> ## References:
>
> [1] Switch Transformers: Scaling to Trillion Parameter Models with Simple and Efficient Sparsity. JMLR 2022.
>
> [2] Multi-modal contrastive learning with LIMoE: the language-image mixture of experts. NeuIPS 2022.
>
> [3] VLMo: Unified Vision-Language Pre-Training with Mixture-of-Modality-Experts. NeuIPS 2022.
>
> ## Q2. Experiments on different tasks and model architectures.
> As suggested by the review comments, we report the experimental results on three different sizes of BERT, including BERT-tiny, BERT-small, BERT-base. To further address your concern, we followed your suggestions and conducted extra experiments on three benchmark tasks (SQuAD, Wikitext and XSum) up on different LMs (including BART-large, GPT-2, T5-base), with the experimental results reported below.
>
> *BART-large on XSum (max_source_length=512, select 2 experts from 8):*
> | | FLOPs | Rouge-2 |
> | :------ | :------: | :------: |
> |Vanilla | 369.3G | 21.9 |
> |MoE | 576.5G	| 22.2 |
> |**MEO**| **369.7G** | **22.4** |
>
>
> *GPT-2 on Wikitext (max_length=1024, select 2 experts from 8):*
> | | FLOPs | PPL |
> | :------ | :------: | :------: |
> | Vanilla | 279.2G | 21.5 |
> | MoE | 395.9G | 21.1 |
> | **MEO** | **279.3G** | **20.9** |
>
>
> *T5-base on SQuAD (max_length=384, select 4 experts from 16):*
> | | FLOPs | Exact_Match |
> | :------ | :------: | :------: |
> | Vanilla | 90.3G | 81.6 |
> | MoE | 221.0G | 82.0 |
> | **MEO** | **90.4G** | **82.1** |
>
> In the new experimental results, our proposed MEO achieved consistent improvement across different tasks. Moreover, MEO’s advantage on computational efficiency still holds when applied to different architectures (i.e., BERT, BART, T5, GPT-2), indicating the universality of our method.
>
> We further evaluate the practical inference speed of Vanilla, MoE and MEO, using T5-base as the base model and “eval_samples_per_second” as the metric. The inference of MEO is significantly faster than that of MoE (176.97 v.s. 106.52), while the speed of Vanilla is 184.7.
>
> ## Q3. Applying MEO to FFN layers.
> Following existing works on MoE, we mainly focus on replacing linear layers in FFN with MoE/MEO layers. In Line 236-251, we have compared MoE layers with and without nonlinearities, and empirically verified that their difference can only cause minor difference in performance, though their outputs differ. Hence, though we discuss less about nonlinear activations in MoE/MEO layers, our method can be applied to routing token / seq to FFN.
>
> ## Q4. The inference speed of MoE and MEO.
> The inference time (when selecting one expert for each input) includes the computation of gating function and allocating specific experts to each input, which costs more than inference on a single model. The total number of the demos is ~10k and the batch size is 16, so the whole process requires more than 600 iterations, which takes a relatively long time. However, further increasing the batch size will make CUDA out of memory, due to the limited resources and memory on the Nvidia V100.
>
> ## Q5. Line 115 - 74G v 7.4G, should be swapped.
> Thank you for pointing out the typo! We will fix it in the next version.

---

### Official Review · Reviewer_6rLC · 2023-08-10

**Soundness:** 2

**Excitement:**

2: Mediocre: This paper makes marginal contributions (vs non-contemporaneous work), so I would rather not see it in the conference.

**Paper Topic And Main Contributions:**

In this paper, the authors introduce a new solution to reduce the computation cost of Mixture of Experts models. Existing MoE systems typically involve high computation overhead on individual experts, to reduce these, the authors propose to merge the experts into one, by reordering the computation. Specifically, the parameters of multiple experts are merged into one. The evaluation results show that the proposed approach significantly improves the computational efficiency of MoE.

**Reasons To Accept:**

1. The authors propose an interesting angle to reduce the computation cost of the experts in sparse MoE models.

2. The authors provide an analysis of the different levels of merging experts into one, as well as the reduced computation.

**Reasons To Reject:**

1. One significant concern raised in this paper is the lack of discussion on the communication overhead of MoE. It's not clear where the performance gain comes from, basically merging experts into one also reduces the communication (all_to_all).

2. The experiment setup is not clear. For example, how many GPUs are used? what is the deployment of the experts?

3. It seems the performance gain is very limited when selecting more experts in table 1, what is the reason?

**Reproducibility:**

3: Could reproduce the results with some difficulty. The settings of parameters are underspecified or subjectively determined; the training/evaluation data are not widely available.

**Reviewer Confidence:**

3: Pretty sure, but there's a chance I missed something. Although I have a good feel for this area in general, I did not carefully check the paper's details, e.g., the math, experimental design, or novelty.

---

> ### Author Rebuttal · Authors · 2023-08-29
>
> Thanks for your detailed comments! We carefully address your concerns as follows:
>
> ## Q1. The difference between MEO and MoE in communication cost.
> There is no significant difference between their required communication costs, though MEO is faster in terms of the inference speed. Both MEO and MoE require distributing inputs to multiple experts and combining the parameters or outputs of selected experts [1, 2].
>
> ## Q2. The experimental setup.
> A detailed experimental setup can be found in Appendix A and Line 220-225. Due to the limited computation resources, we use a NVIDIA V100 to conduct each experiment. Our deployment of experts is identical to previous works and uses their used pytorch packages to allocate experts in parallel.
>
> ## Q3. Performance of selecting more experts.
> The results in Table 1 demonstrate the advantage of selecting multiple experts (when compared with selecting only one expert) but selecting excessive experts degrades the performance to be suboptimal. This is consistent with previous work [3] results reported in their Appendix. The suboptimal performance, as discussed in Line 113-116, could result from the interference between experts [4, 5]. Moreover, selecting excessive experts reduces the number of possible combinations of selected experts (e.g., 1 when selecting 32 from 32 v.s. 35960 when selecting 4 from 32).
>
> ## References:
> [1] Sparse MoE with Random Routing as the New Dropout: Training Bigger and Self-Scalable Models. ICLR 2023.
>
> [2] Parameter-Efficient Mixture-of-Experts Architecture for Pre-trained Language Models. COLING  2022.
>
> [3] Outrageously Large Neural Networks: The Sparsely-Gated Mixture-of-Experts Layer. ICLR 2017.
>
> [4] Multimodal Contrastive Learning with LIMoE: the Language-Image Mixture of Experts. NeuIPS 2022.
>
> [5] Uni-Perceiver-MoE: Learning Sparse Generalist Models with Conditional MoEs. NeuIPS 2022.

---

### Meta-Review · Area_Chair_LDAr · 2023-09-17

**Recommendation:** 5

**Metareview:**

The paper proposes to merge experts of mixture of experts models into one expert during inference, which improves inference efficiency and task performance

**Pros**:
- the proposed expert merging technique is simple drop-in replacement to MOE models to effectively improve inference efficiency and task performance.
- The authors provide additional positive results on larger-scale models and on more generative tasks to demonstrate the effectiveness of the approach.

**Cons**: I find that the work doesn't exhibit notable drawbacks, and the concerns and questions raised in the reviews have been adequately resolved.

---

### Decision · Program_Chairs · 2023-10-07

**Decision:**

Accept-Main

**Comment:**

The paper proposes to merge experts of mixture of experts models into one expert during inference, which improves inference efficiency and task performance

**Pros**:
- the proposed expert merging technique is simple drop-in replacement to MOE models to effectively improve inference efficiency and task performance.
- The authors provide additional positive results on larger-scale models and on more generative tasks to demonstrate the effectiveness of the approach.

**Cons**: I find that the work doesn't exhibit notable drawbacks, and the concerns and questions raised in the reviews have been adequately resolved.